# Enhancing the Biological Oxidation of H_2_S in a Sewer Pipe with Highly Conductive Concrete and Electricity-Producing Bacteria

**DOI:** 10.3390/ijerph20021459

**Published:** 2023-01-13

**Authors:** Huy Thanh Vo, Tsuyoshi Imai, Masato Fukushima, Kanathip Promnuan, Tasuma Suzuki, Hiraku Sakuma, Takashi Hitomi, Yung-Tse Hung

**Affiliations:** 1Faculty of Urban Engineering, Mientrung University of Civil Engineering, Tuy Hoa 620000, Vietnam; 2Graduate School of Science and Technology for Innovation, Yamaguchi University, Yamaguchi 7558611, Japan; 3Department of Biotechnology, Faculty of Technology, Khon Kaen University, Khon Kaen 40002, Thailand; 4Nagasaki Humepipe Industry Co., Ltd., Ibaraki 3000051, Japan; 5Department of Civil and Environmental Engineering, Cleveland State University, Cleveland, OH 44115, USA

**Keywords:** hydrogen sulfide, sewer pipe, conductive concrete, Denka Black, electricity-producing bacteria, San-earth

## Abstract

Hydrogen sulfide (H_2_S) generated in sewer systems is problematic to public health and the environment, owing to its corrosive consequences, odor concerns, and poison control issues. In a previous work, conductive concrete, based on amorphous carbon with a mechanism that operates as a microbial fuel cell was investigated. The objective of the present study is to develop additional materials for highly conductive concrete, to mitigate the concentration of H_2_S in sewer pipes. Adsorption experiments were conducted to elucidate the role of the H_2_S reduction. Additionally, electricity-producing bacteria (EPB), isolated from a municipal wastewater treatment plant, were inoculated to improve the H_2_S reduction. The experimental results showed that inoculation with EPB could decrease the concentration of H_2_S, indicating that H_2_S was biologically oxidized by EPB. Several types of new materials containing acetylene black, or magnetite were discovered for use as conductive concrete, and their abilities to enhance the biological oxidation of H_2_S were evaluated. These conductive concretes were more effective than the commercial conductive concrete, based on amorphous carbon, in decreasing the H_2_S concentration in sewer pipes.

## 1. Introduction

The damage to sewer concretes by sulfide-induced corrosion is increasing and thus poses new challenges for sustainable urban management. The growing sanitary infrastructure, including sewer pipeline networks and effluent wastewater systems, is an undesirable trend for managers who must increasingly accommodate a larger portion of their finances for their operation and maintenance. Hydrogen sulfide (H_2_S) generated in sewer systems becomes problematic to public health and the environment, owing to its corrosive consequences, odor concerns, and poison control issues [1]. Sulfide-induced corrosion of sewer concrete causes a loss of material bulk, rupturing of the pipe frame, and, eventually, the structural destruction. Governments worldwide spend millions to billions of dollars annually on the maintenance, rehabilitation, and repair of sewer systems. A report published in 2021 by the American Society of Civil Engineers, estimated a cost of more than USD 3 billion for the national sewer rehabilitation in 2019, to replace approximately 7560 km of pipeline nationwide. In Germany, 40% of the damage of underground pipelines in private and public sewage systems was determined to have resulted from biogenic corrosion, and the necessary repair and maintenance were estimated to be approximately USD 40 billion [2]. The annual repair costs of the sewer pipeline infrastructures in Australia and New Zealand are valued in further stages to be AUD 80 and NZD 16 billion, respectively [3]. In the past decade, China has invested nearly USD 19 billion in the municipal sewer infrastructure. However, a major problem with this developing pipeline network is the pipe corrosion by biogenic sulfuric acid attacks [4]. In Japan, the number of pipeline facilities that have exceeded their service life is increasing rapidly over time, and local authorities must formulate plans for the maintenance and renewal of the existing pipeline systems, while reducing the life-cycle costs. Thus, the diminution, eradication, or control of H_2_S should be the goal of all civil engineers in the field of sewer construction.

Several studies on the biocorrosion of concrete sewers have heightened the understanding of the mechanism of concrete deterioration by H_2_S [3,5,6,7,8,9,10]. There has been rapid advancement in the field of construction technology in the last three decades, to overcome the challenges and obstacles caused by H_2_S biogenic corrosion. To date, research has mainly focus on the H_2_S elimination rather than the inhibition of the H_2_S generation. Consequently, the studies have focused on chemical oxidation solutions to suppress the corrosion by biogenic sulfuric acid attacks. It can be assumed that the H_2_S removal requires chemical oxidants or alternative electron acceptors. According to the United States Environmental Protection Agency (US EPA) [1], a simple procedure to eliminate the generated H_2_S from the anaerobic slime layer of sewer pipes is to design the optimal gradient and flow velocity to prevent the trapping of grit and organic sediments. Despite their efficacies, these approaches suffer from several major drawbacks when applied to real conditions on a larger scale. There are similarities between the H_2_S removal by the oxygen injection and air pressurization, as described by Gutierrez (2008) and Mohanakrishnan (2008). These methods have the advantages of organic matter obliteration, acidic gas control, and environmentally friendly treatment [11,12,13,14]. Air injection is easily used in sewer systems but interferes with the efficiency of the oxygen transfer [15,16]. Pressurized air allows for a better dissolution of oxygen into the sediment layer; however, there is a potential fire hazard. A recent study by Zhang et al. (2022) improved upon the previous studies by using air-nanobubbles for odor control and sewer corrosion. The effect of the oxygen supply increased by 5.04 times, compared with the traditional methods [17]. The use of advanced oxygen processes or chemical oxidants is effective in suppressing H_2_S in a short time and has a good oxidation potential. A variety of chlorines that are used as disinfectants are highly reactive and readily oxidize sulfur to sulfides. However, the residual chlorines, even at low concentrations, are hazardous to the aquatic environments. Devai and Delaune (2002) proposed potassium permanganate to be the most effective chemical, followed by hydrogen peroxide and sodium hypochlorite, in reducing the sulfide products [18]. However, the high cost and undesirable by-products limit their use under real conditions. In 2010, Can-Dogen et al. demonstrated that the addition of nitrate as an electron acceptor could significantly reduce the sulfide concentration and hydraulic retention time in effluent streams [19]. 

Several studies have shown that in addition to the chemical oxidation that suppresses the H_2_S emissions in sewer pipelines, the inhibition of the sulfate-reducing bacteria (SRB) activity also contributes to a low H_2_S-related pipe damage. Another approach to control H_2_S is the pH elevation shock for the deactivation of anaerobic bacteria. The supplementation of magnesium hydroxide or sodium hydroxide to increase the pH to 9–12 in a rapid contact time, can inhibit SRB in the slime layer by 50% with a long effective period (up to a few weeks) and prevent H_2_S from forming a gas phase [20]. This is consistent with other observations, which showed that molybdite, formaldehyde, and nitrite have a beneficial bactericidal effect in preventing the sulfate reduction and controlling sulfide emissions [13,21,22]. Jiang and Yuan (2013) used a combination of free nitrous acid and hydrogen peroxide to inhibit the microbes of the anaerobic biofilm and achieved a 99% biocidal effect when the biofilms were exposed to 0.2 mg N/L (nitrous acid) and 30 mg/L (H_2_O_2_) for 6 h or longer [23]. However, using these compounds could negatively affect the environment under real sewer system conditions. 

However, the aforementioned studies conducted to mitigate the H_2_S emersion and corrosion in sewers disregard the fact that there are many long-standing pipelines. The corrosion problem of sewer systems in a state of deterioration and collapse remains a concern for global authorities under the budget and operating maintenance pressures. Further investigations are thus required to establish the feasibility of real sewer infrastructure and a more economic efficiency. 

To overcome the limitations of previous studies, research was conducted to provide preventive solutions to obtain a more sustainable material, to combat the biocorrosion of concrete sewers and to minimize the H_2_S formation in the anaerobic sediment. Previously, conductive concrete, based on amorphous carbon, which operates as a microbial fuel cell (MFC), was developed. This method employs movement through the electronic pathway from the sediment layer to the surface water through the conductive concrete material. Thus, SRB can be easily activated, significantly decreasing the biogenic corrosion issues. Herein, the mechanism of conductive concrete using different materials and the contributions of electricity-producing bacteria (EPB) to the reduction of generated sulfide were elucidated. First, the mechanism by which the commercially used conductive materials suppress the H_2_S inhibition was investigated. Then, trials using new conductive materials, combined with the EPB inoculation were conducted to verify the biological oxidation of H_2_S reduction. Finally, the new conductive concrete was assessed to determine the effect of the H_2_S reduction.

## 2. Materials and Methods

### 2.1. Oxidation of Sulfide by the Electron-Emitting Bacteria

H_2_S is generated by the sulfate-reducing bacteria in the anaerobic environment of the biofilm layer. Previous work has concentrated on elucidating the H_2_S reduction in three ways: adsorption on amorphous carbon, chemical oxidation, and biological processes. One possible explanation for this is the combination of the adsorption capacity of conductive substances and the biological oxidation. An experimental scheme in which an electronic pathway is blocked inside the conductive concrete has no effect on the sulfide reduction. The only way to reduce H_2_S formed in the conductive concrete is via biological oxidation. 

Therefore, in this study, the work plan assumed the oxidation of sulfide by electron-releasing bacteria, as shown in Figure 1. First, the H_2_S removal occurs in the anodic section of the conductive concrete surface of the biofilm layer, the EPB activity releases electrons accompanied by the H_2_S oxidation of the organic matter. Subsequently, the electrons transfer to the electron acceptors through the electronic pathway inside the conductive concrete. Finally, at the liquid–air interface, oxygen, as an electron acceptor, reacts to produce water.

The aforementioned methods for testing the adsorption capacity of the different conductive materials were examined, and the effect of the H_2_S reduction was assessed. The experiments involving the inoculation of EPB were conducted, and the results were compared to investigate the efficiency of the conductive concrete.

### 2.2. Experimental Models and Materials 

All experiments were performed at room temperature (24 °C ± 0.1) using distilled water (SA-2100A; A type, Tokyo Rika Kikai Co., Ltd., Tokyo, Japan). The reagents used were of special grade obtained from Kishida Chemical Co., Ltd., Nacalai Tesque Co., Ltd., Osaka, Japan and FUJIFILM Wako Chemicals Industries, Ltd., Tokyo, Japan. The materials used as conductive concrete (Figure 2) and their basic properties are described below.
-San-earth M5C, commercially manufactured by Sankosha Co., Ltd., Tokyo, Japan (abbreviated as S), contained amorphous carbon produced from oil refining as a conductive component, with a maximum size of 0.3 mm (a specific gravity of approximately 1.285 g/cm^3^ and surface area of 1.9 m^2^/g). Its content was approximately 50 wt.% (amounts of conductive substance in the total powder) (S50), with a water-to-powder ratio of 42%.-Denka Black (trade name of Denka Acetylene Black) was manufactured by Denka Co., Ltd., Tokyo, Japan (abbreviated as D), with a mean size of 35 nm (a specific gravity of approximately 0.04 g/cm^3^ and surface area of 50–68 m^2^/g). Its contents were approximately 5 wt.% D5 and 10 wt.% D10, with water-to-powder ratios of 50% D5 and 62% D10.-Magnetite MTB-30 (trade name Magnetite) was manufactured by Morishita Bengara Kogyo Co., Ltd., Iga City, Japan (abbreviated as M), with a mean size of 0.5 µm (surface area of 13.3 m^2^/g). Its contents were approximately 5 wt.% M5, 25 wt.% M25, and 50 wt.% M50, with water-to-powder ratios of 38% M25 and 40% M50.

Ordinary Portland cement (OPC), manufactured by Sumitomo Osaka Cement Co., Ltd., Tokyo, Japan, was used as the normal concrete. This OPC had a specific gravity of 3.15 g/cm^3^ and a specific surface area of 0.3350 m^2^/g. Subsequently, all concrete samples without aggregates were cured and treated with an alum-based scouring removal agent.

### 2.3. Preparation of the Cup-Shaped Concrete Specimens

The geometry of the cup-shaped specimens prepared using OPC, San-earth, and the new conductive concretes (Denka Black and Magnetite MTB-30) is shown in Figure 3. The cup-shaped specimen has a capacity of approximately 1.9 L. The specimens prepared in this study are called “cement paste” because they contain no aggregate but are referred to as “concrete” for convenience, including in future practical applications [24]. The research methodology is illustrated in Figure 3.

### 2.4. Measurement of the Electrical Resistivity 

The electrical resistivities of the new conductive concrete materials were measured using a chemical impedance analyzer (model: IM3590) manufactured by Hioki Co., Ltd., Nagano, Japan, based on the widely used four-electrode method [25]. 

These specimens were cured in water for 27 days, considering that the actual conduits are wet. Because concrete contains several electrolytes, the electrical resistivity changes with the application time. The electrical resistivities reported in this study were the values after 5 min, which were confirmed by the preliminary experiments that became stable, even in the low-frequency region, where it takes a longer time to converge to a constant value. The applied voltage and power frequency during the measurements were set to 0.3 V and 50 Hz, respectively. 

### 2.5. Sulfide Adsorption Experiments

To confirm that the concrete materials do not contribute significantly to the effect of the sulfide reduction, San-earth (S50), Denka Black (D10), and magnetite (M50) tests were performed to determine the adsorption capacity. Three vials containing 100 mL of distilled water with a dissolved oxygen concentration of less than 0.1 mg O_2_/L by nitrogen bubbling, were prepared for each conductive material, and Na_2_S.9H_2_O was added to reach a concentration of 20.5 mg S/L. The pH was maintained in the range of 7.0 ± 0.1. The adsorption apparatus was immediately sealed with a butyl rubber plug and then shaken. Samples were collected after a predetermined period to measure the sulfide ion concentration over time, considering that the sulfide oxidation during the sample preparation was negligible. All experiments were performed in triplicates. Sulfide was quantified by colorimeter, using the reagents to form methylene blue, according to the US EPA method (376.2).

### 2.6. Preparation and Cultivation of EPB for the Trials of the Sulfide Reduction with Conductive Concrete

#### 2.6.1. Culture of EPB Isolated from the Sewer Sludge

The excess and digested sludge collected from the Ube City Wastewater Treatment Plant were isolated and cultivated in Luria–Bertani broth (Wako Chemical Co., Ltd., Tokyo, Japan) at 30 °C. DNA was extracted using a Mora-Extract kit (Kyokuto Pharmaceutical Industrial Co., Ltd., Tokyo, Japan), according to the instructions of the manufacturer. A polymerase chain reaction was performed with the touchdown method using primers GC-341f and 518r for the amplification of the 200 bp 16S rDNA, which were analyzed using a DNA sequencer Ion S5 (Thermo Fisher Scientific Co., Ltd., Waltham, MA, USA) [26,27]. The species results were visualized using Bio Edit software (SD Biosystems Co., Ltd.) and the National Center for Biotechnology Information website. As a result, two species, *Morganella morganii* and *Proteus alimentorum* were identified from the excess sludge, and two species, *Proteus mirabilis* and *Shigella sonnei* were identified from the digested sludge with a homology of more than 99.7%. These are typical intestinal bacteria, while those other than *P. alimentorum* are EPBs [27,28,29,30]. Related to the sequence of *P. alimentorum,* this belongs to the *Proteus* genus (including *P. mirabilis)* and is also an EPB [28,31,32]. 

#### 2.6.2. Preparation of the Samples of EPB for the Sulfide Suppression by Conductive Concrete

Two samples of OPC and San-earth were prepared to assess the contribution of EPB in the H_2_S reduction. Each specimen was filled with 0.14 L of the digested sludge—suspended solids (SSs) concentration of 17,160 mg/L and volatile suspended solids (VSSs) concentration of 12,800 mg/L—thoroughly mixed with 1.12 L of synthetic wastewater.

The synthetic wastewater containing the following compositions was prepared, as described by Zhou (2006)—(g/L distilled water): 2 glucose, 0.85 NH_4_Cl, 2 mM K_2_HPO_4_, 2 NaHCO_3_, 0.02 yeast extract, 0.7 (NH_4_)_2_HPO_4_, 0.75 KCl, 0.42 FeCl_3_, 6H_2_O, 0.81 MgCl_2_, 6H_2_O, 0.25 MgSO_4_, 7H_2_O, 0.15 CaCl_2_, 6H_2_O, and 0.018 CoCl_2_, 6H_2_O [33]. 

Other experiments were similarly set up with OPC and San-earth specimens. Microbial suspensions of *M. morganii*, *P. alimentorum*, *P. mirabilis,* and *S. sonnei* were added, each at approximately 10^8^ CFU/mL. The experiment was started with the stabilized mixture, and the performance of the sulfide reduction was evaluated, based on the measurements of pH, sulfate ions, and sulfide concentrations over time. The pH and dissolved oxygen were measured using a pH meter (D-72) and DO meter (OM-71-2), respectively (HORIBA Advanced Techno Co., Ltd., Kyoto, Japan). Magnesium sulfate and glucose were added on days 5, 10, and 15.

The sulfate concentration in the water samples was measured using the barium sulfate turbidimetric method, according to the US EPA method (375.4) after filtration through a 0.45-µm membrane filter.

### 2.7. H_2_S Suppression Experiment by Conductive Concrete

The subjects of the different conductive concretes were asked to pay close attention, to evaluate the effect of the sulfide reduction over a long-term period. The specimens were used and filled with 0.14 L of the sludge mixture (a volume ratio of 1:1), comprising the excess (SS concentration of 6080 mg/L and VSS concentration of 5210 mg/L) and digested sludge (SS concentration of 12,820 mg/L, VSS concentration of 9820 mg/L), thoroughly mixed with 1.12 L of artificial wastewater (as described above). The experiment was started with the stabilized mixture, and the performance of the sulfide reduction was evaluated, based on the measurements of pH, sulfate ions, and sulfide concentrations over time. Once the results confirmed that sulfate was nearly completely consumed by SRB, a substrate of glucose (100 mg/L) and magnesium sulfate (33 mg S/L) was instantaneously added.

## 3. Results

### 3.1. Measurement of the Electrical Resistivity

The results obtained from the preliminary measurements of the electrical resistivity of various conductive concretes are presented in Figure 4. OPC has the strongest resistance to the flow of electrons, having the electrical resistivity of approximately 18 Ω·m. A negative correlation was found between the amounts of Denka Black and Magnetite MTB-30, in the total powder and electrical resistivity. Increasing the weight ratio of the conductive substances in concrete resulted in a lower electrical resistivity. Denka Black has a resistivity of more than 8.4 Ω·m with a content of 5 wt.% (D5) and was reduced to approximately 5.6 Ω·m at a weight content of 10% (D10). From the magnetite results, three samples were collected, magnetite was added gradually at 5% (M5), 25% (M25), and 50% (M50) weight content, and the electrical resistivity decreased from more than 11 Ω·m to 6.6 and 5.3 Ω·m, respectively. An individual sample of San-earth had a 50 wt.% (S50) stated outcome of nearly 6.5 Ω·m, whereas that of Denka Black was only 10 wt.% (D10).

Denka Black improved the concrete conductivity more effectively than magnetite and San-earth. M50 had the lowest electricity resistivity, which is approximately the same as that of D10 (only 10% weight content of Denka Black). M50 and S50 had the same conductive substance content, and the resistivity of San-earth was higher than that of magnetite. There was a significant difference between the conductivity effect of the new conductive materials used in this study and that of the commercially conductive concrete of San-earth.

### 3.2. Results of the Adsorption Experiment of H_2_S

In the sulfide adsorption experiments, the most conductive materials, Denka Black (D10), San-earth (S50), and magnetite (M50), were chosen for the H_2_S adsorption experiment. The equilibrium was achieved after approximately 8 h.

Figure 5 shows that Denka Black (D10) was the least effective in the H_2_S adsorption, followed by San-earth and Magnetite (M50), which were considered the best adsorption materials. 

### 3.3. Effect of the EPB Inoculation on the H_2_S Reduction

To assess the role of the EPB activity in suppressing the efficiency of the H_2_S, specimens of OPC and San-earth, with and without the EPB inoculation, were used to evaluate the pH, and sulfate, and sulfide concentrations for 20 days. Figure 6 shows the intercorrelations of these parameters with time. The histogram in Figure 6A shows the pH change. At the initial stage of the substrate addition, the pH increased to approximately 9 because of the addition of glucose, and then the pH of all samples decreased rapidly to approximately 4.5, which can be attributed to the generation of organic acids. The pH started increasing slightly when the organic acids were transformed to volatile organic compounds associated with nitrogen-containing organic compounds that decomposed to cause the alkalinity and elution of calcium hydroxide in the concrete materials. 

As shown in Figure 6B, no significant difference in the sulfate transformation was observed with the change in pH. Sulfate ions decreased significantly within a few days in all cases of the magnesium sulfate addition, and this transition was determined to be independent of the type of concrete, with or without the EPB inoculation.

Figure 6C shows strong evidence of the influence of the EPB inoculation on the experimental models. For the different concretes with and without the EPB inoculation, the mean transformation of sulfide was differentiated. The sulfide ions of the San-earth specimen exhibited lower concentrations than those of the OPC. This is consistent with Figure 4, where the conductive substance in the San-earth is significantly higher than that in the OPC. Particularly, the sulfide concentration decreases significantly in the San-earth specimens inoculated with the EPB communities, compared with those not inoculated. 

To evaluate this effect quantitatively and more intuitively, the average sulfide concentration over the 20-day experimental period was calculated, and the reduction percentages of sulfide in the EPB strains inoculated with OPC and San-earth with and without the inoculation of EPB were compared, as shown in Figure 7. San-earth with the EPB inoculation had the highest percentage of the H_2_S reduction (69%), whereas OPC with the inoculation and Sun-earth without the inoculation had percentages of 41% and 37%, respectively.

Figure 7 demonstrates that the addition of microorganisms of EPB positively influences the biological oxidation of H_2_S in the specimen material, especially the conductive concrete. 

### 3.4. H_2_S Suppression Using Highly Conductive Concrete

The results indicated that concrete with a high density of conductive substances and EPB communities significantly affects the H_2_S inhibition. In this experiment, traditional concrete (OPC), commercial conductive concrete (San-earth, S50), and a highly conductive material (D10) were considered, in association with the EPB inoculation, to assess the advanced capacity of the H_2_S inhibition.

The results obtained from monitoring the pH, sulfate, and sulfide concentrations for 134 days are presented in Figure 8. The progress of the pH and sulfate ions over time followed the same trend, as that shown in Figure 6. The sulfide change (Figure 8C) from the data comparison of OPC and San-earth, demonstrates that the new conductive concrete, Denka black (D10), was the most effective in reducing H_2_S, and a representative example is shown. 

The reduction percentage of the sulfides relative to OPC, calculated from the average sulfide concentration after 134 days, is shown in Figure 9. 

The column chart shows a few characteristics of concrete. D5 and M5 did not exhibit a H_2_S reduction. When the weight content of the conductive substances of Denka Black, magnetite, D10, M25, and M50 reached a high percentage of the H_2_S inhibition (nearly 75%, 32%, and 64%, respectively). The reduction rate for San-earth (S50) was more than 40%.

## 4. Discussion

The aim of this study was to assess the importance of conductive concrete in facilitating the biological oxidation to reduce H_2_S by electricity-generating bacterial communities in effluent wastewater. Previous studies used a temperature of approximately 24–30 °C, a hydraulic retention time, and pH, as those used in this experiment, which were proposed to be appropriate for activating the MFCs [33,34,35]. Electrochemically dynamic bacteria were used to support the biofilm layer attached to the concrete. This support accelerated the biological oxidation of H_2_S at the root of the conductive concrete. For the EPB-inoculated specimens, the OPC-inoculated specimens with a high resistivity resulted in a reduction in H_2_S (Figure 6). However, a significant difference was observed between the inoculation and no inoculation of the conductive San-earth material. This could be attributed to the presence of the activated EPB that could oxidize the sulfide matter in the OPC specimen, and the accumulation of electrons formed in the process of the biological oxidation. Moreover, under anaerobic conditions, oxygen is not available to accept the available electrons. Biological reactions, at later stages, were either affected significantly or the effects were limited. Therefore, connections might exist between the EPB communities and the electronic pathway on the concrete, to enhance the biological oxidation for the effective H_2_S suppression. Practically, there are many electricity-generating bacteria in wastewater and sewer systems [36,37,38,39]. The abovementioned processes occur spontaneously, therefore, it may be necessary to use a conductive concrete to provide a stable environment for EPB to work effectively, following the described mechanism.

Previous findings failed to provide a clear explanation of the adsorption capacity and biological oxidation, particularly regarding the H_2_S inhibition [24]. Moreover, the adsorption capacity of the new conductive concretes for the H_2_S reduction (Figure 5) is concerning. These materials are representative of the most conductive substances with conductivities (Figure 4): M50 > D10 > S50. Magnetite (M50) and Denka Black (D10) were found to be more effective for absorbing H_2_S than the commercial San-earth. However, no evidence was found that the adsorption capacity of M50 was more effective in the H_2_S reduction, as illustrated in Figure 9. Accordingly, a sample of D10 with a Denka Black content of 10 wt.% had a significant H_2_S reduction (approximately 75%), suggesting that the adsorption of a material does not contribute to the main mechanism for eliminating H_2_S, even though its capacity is reduced for the sulfide reduction. 

Conductive concrete did not reduce sulfides when the conductivity was less than 0.12 (1/(Ω-m)), but almost all materials with higher conductivities revealed an evident deep reduction of H_2_S. A linear correlation (Figure 9) was obtained from the above relationship when all conductivities were higher than 0.12 (1/(Ω-m)). Although the exclusion of the inverse of the electrical resistivity, at less than 0.12 (1/(Ω-m) did not reduce sulfides, these results should be cautiously interpreted. Consequently, the D10 and M50 specimens were again confirmed to be the most effective. During the experiment (134 days), the conductive concrete with a low concentration of H_2_S in all samples (lower than 0.5 mg/L) was observed (S50: 50%, M50: 38%, and D10: 58%). Therefore, on using conductive concrete in sewer systems, sulfide ions may often exist at lower levels and pose a lower risk of H_2_S-related pipe damage. Although, these findings should be scrutinized, the inoculation of the EPB communities significantly accelerated the sulfide oxidation when highly conductive concrete was used. Further studies on this topic are therefore needed to select a conductive material that is less expensive and more effective in reducing sulfides. Furthermore, the transfer electrons via the electronic pathway inside the conductive material to oxygen as electron acceptors at the water–air interface, is crucial. Additionally, the convexity or concavity of the surface of the material must be considered as a factor in decreasing the sulfide reduction.

## 5. Conclusions

The findings of this study provide a better understanding of the mechanism of the sulfide reduction by conductive concrete in sewage pipes. The current investigation determined a highly conductive concrete that showed a higher sulfide reduction than that achieved with the commercially manufactured conductive concrete. Specifically, the results for the conductive materials using Denka Black of 10 wt.% (D10) and magnetite of 50 wt.% (M50) helped clarify the role of highly conductive concrete, in comparison with the San-earth (S50) sample. Although a lower contribution of the adsorption process from the conductive materials to the effect of the H_2_S suppression was found, this study partially substantiated that the inoculation with EPB enables the biological oxidation to reduce the sulfide generation significantly. Further studies using the same experimental setup are needed, including the measurement of the compressive strength of the material, estimation of the manufacturing cost, trials to use less expensive conductive materials with higher sulfide reduction effects, based on the mechanism determined in this study, or a long-term demonstration study using real sewage water.

## Figures and Tables

**Figure 1 ijerph-20-01459-f001:**
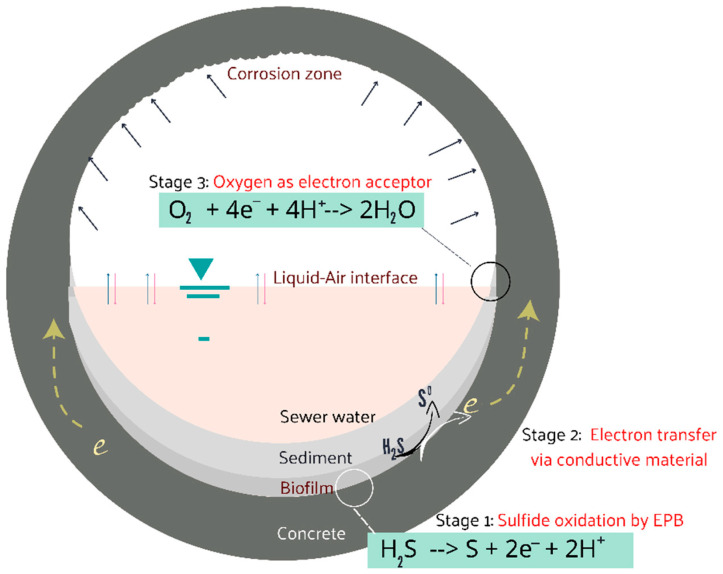
Mechanism of the sulfide oxidation by the electron-emitting bacteria.

**Figure 2 ijerph-20-01459-f002:**
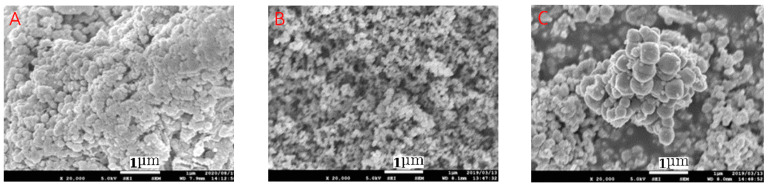
Scanning electron microscopy (SEM) images of the prepared conductive materials: (**A**) San-earth M5C, (**B**) Denka Black, and (**C**) Magnetite MTB-30.

**Figure 3 ijerph-20-01459-f003:**
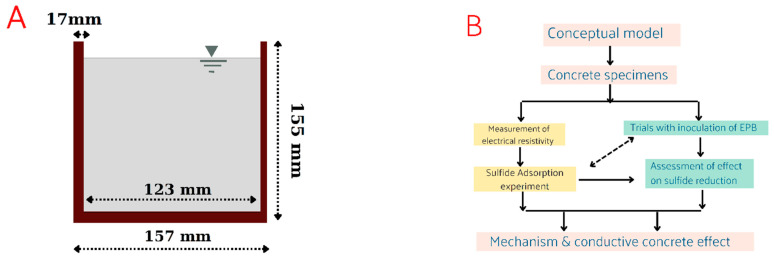
Geometry of the cup-shaped specimen produced (**A**) and the research methodology flowchart (**B**).

**Figure 4 ijerph-20-01459-f004:**
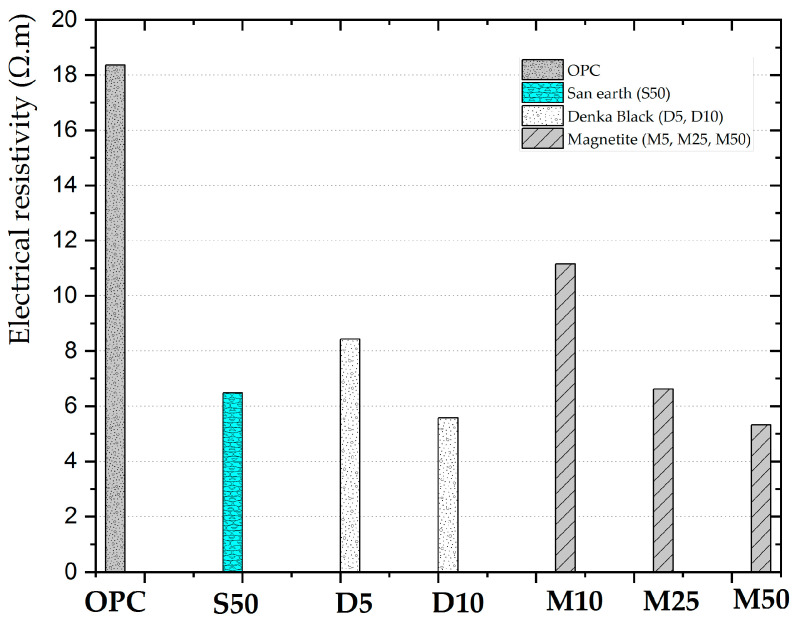
Electrical resistivity of the concrete used in this study at 50 Hz.

**Figure 5 ijerph-20-01459-f005:**
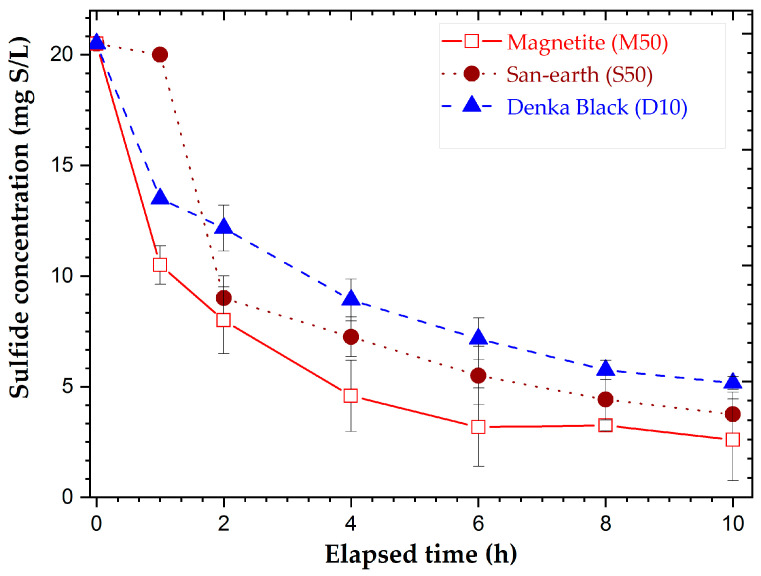
Effect of the sulfide adsorption with time for three different materials.

**Figure 6 ijerph-20-01459-f006:**
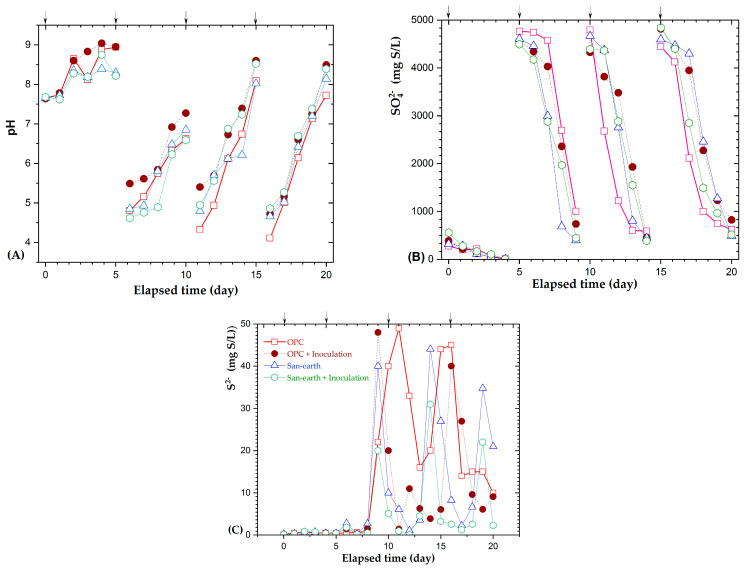
Effects of the EPB inoculation on the pH (**A**) and sulfate (**B**), and sulfide concentrations (**C**). The arrows (↓) indicate the addition of glucose and magnesium sulfate; 1000 mg/L of glucose and 500 mg S/L of sulfate ion were added at the start of the experiment, and 10,000 mg/L of glucose and 5000 mg S/L of sulfate ion were added on days 5, 10, and 15.

**Figure 7 ijerph-20-01459-f007:**
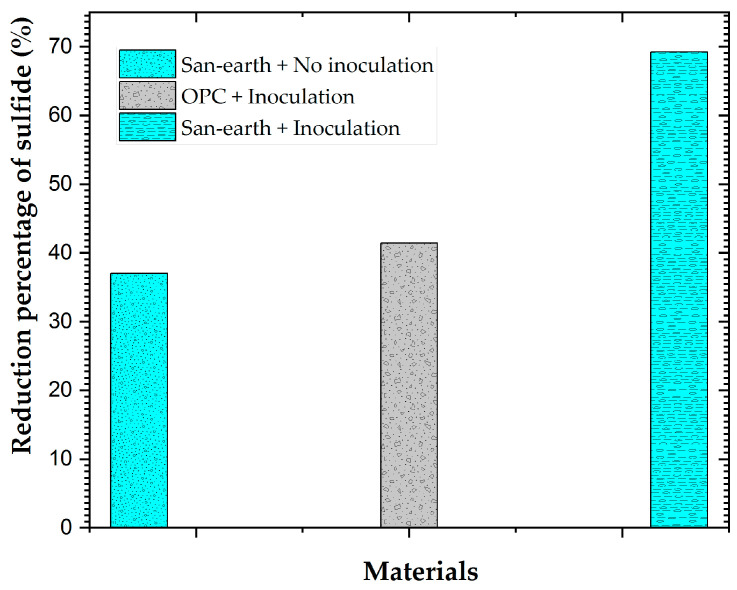
Percentage of the sulfide reduction in the experiments with/without the inoculation with EPB.

**Figure 8 ijerph-20-01459-f008:**
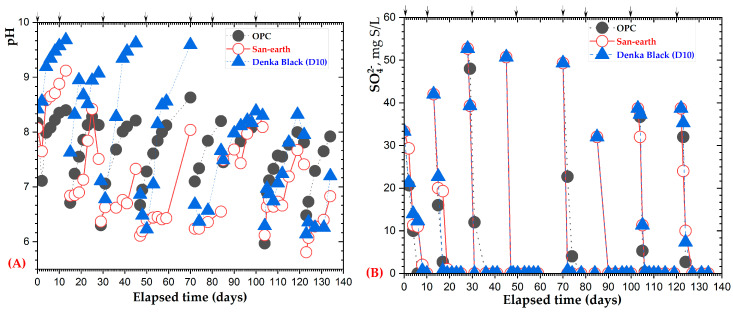
Sulfide reduction experiments using concrete: OPC, San-earth, and Denka Black. The arrows indicate the addition of glucose and magnesium sulfate. (**A**) pH, (**B**) sulfate, and (**C**) sulfide concentrations. Arrow (↓) indicates the addition of glucose and magnesium sulfate.

**Figure 9 ijerph-20-01459-f009:**
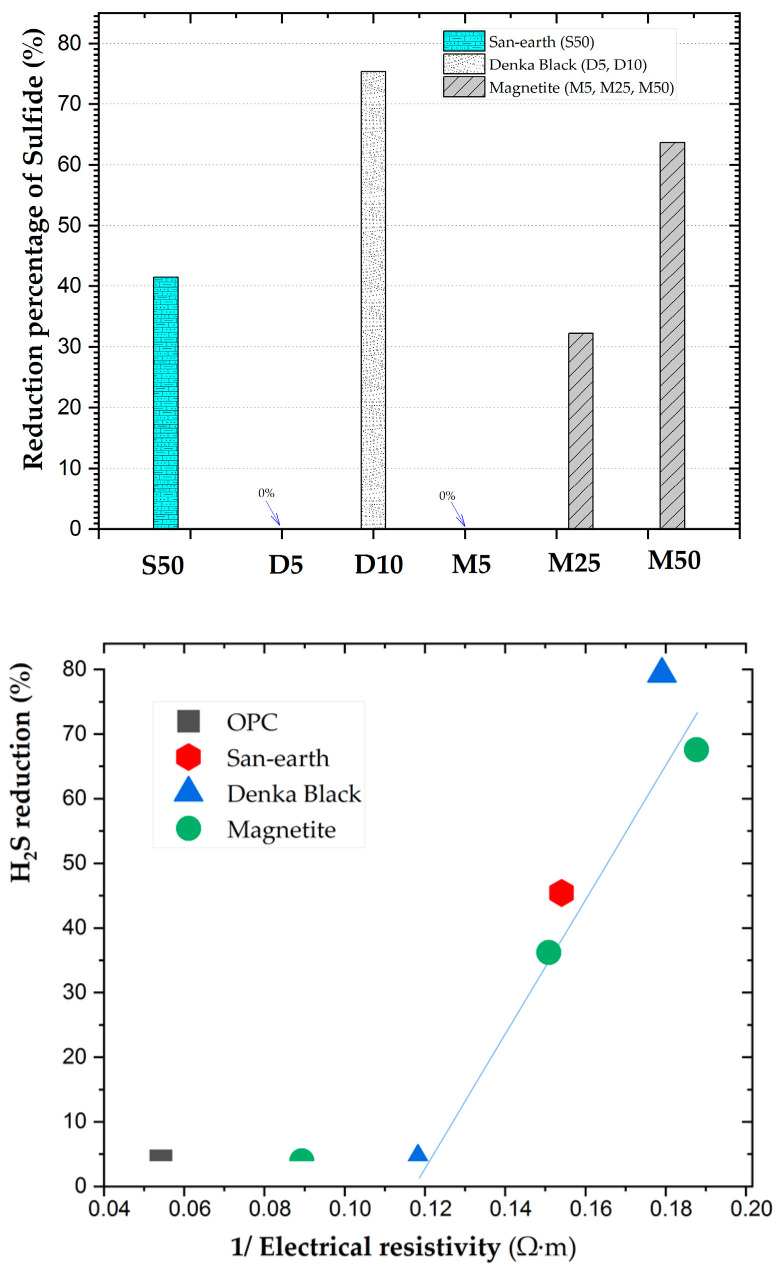
Percentage of the sulfide reduction obtained for the different conductive concretes; relationship between the concrete conductivity and the sulfide reduction.

## Data Availability

Not applicable.

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
