# Peer review of "Enhancing the Biological Oxidation of H2S in a Sewer Pipe with Highly Conductive Concrete and Electricity-Producing Bacteria"

_ijerph, 2023, doi:10.3390/ijerph20021459_

Round 1

Reviewer 1 Report

Microbiologically induced concrete corrosion (MICC) is a specific occurrence in sewer systems where the cementitious materials are eroded toward a paste by microbiological processes. MICC has been one of the factors causing huge asset losses and urban hazards worldwide. The main cause of MICC is the involvement of a biologically induced sulfur cycle process:

Firstly, in an anaerobic environment, which usually occurs in the sediment, SRB reduces sulfate in the effluent sediments to H2S; then H2S overflows the effluent and adsorbs on the concrete surface above the water level, and is further converted to H2SO4 by SOB. The production of sulfuric acid means the beginning of concrete corrosion.

Reducing the concentration of hydrogen sulfide is one of the means to inhibit corrosion. The authors performed a valuable exploration by eliminating the concentration of H2S through electricity-producing bacteria. With this aim, they prepared concrete with different electrical conductivity, whereby they explored the contribution of adsorption in reducing hydrogen sulfide on the one hand, and the contribution of inoculated EPB in reducing hydrogen sulfide on the other hand. But there are many aspects of the article that need to be improved. My comments are as follows:

(1)       From the introduction section, the practical significance of this study is to reduce the corrosion problem in the wastewater system, however, the causes of corrosion in the wastewater system are not introduced. This aspect is what needs to be added. It is recommended to read the following recent articles (or some others. If needed, please search by yourselves)

(a). Madraszewski S, Dehn F, Gerlach J, et al. Experimentally driven evaluation methods of concrete sewers biodeterioration on laboratory-scale: A critical review[J]. Construction and Building Materials, 2022, 320: 126236.

(b). Wu M, Wang T, Wu K, et al. Microbiologically induced corrosion of concrete in sewer structures: A review of the mechanisms and phenomena[J]. Construction and Building Materials, 2020, 239: 117813.

(2) In Figure 1 and section 2.1 Oxidation of sulfide by electron-emitting,

(a)the biofilm is presented to be below the sediment, which is not quite realistic. There is no clear boundary between the two, anaerobic bacteria are in presence of the sediment. Please redraw it.

(b) Why do electrons pass through concrete when they can be transferred directly through sewage? This is not following the laws of physics, because the resistance of concrete is much higher than that of water. Please clarify and explain this aspect

(c) Line 129, The authors stated that “Therefore, in this present study, the work plan designed assumed that "oxidation of sulfide by electron-releasing bacteria" as shown in Figure 1. ”  Why is it based on a hypothesis, for experimental verification? Is it to match the evidence? Please provide relevant supporting research.

(d) After H2S is generated, there are two paths available to it, path 1 is to diffuse into the effluent and have the opportunity to spill over the water level causing concrete corrosion, and path 2 is, as the authors assumed, to be adsorbed onto the concrete surface, meaning that they are converted into elemental sulfur in the sediment. Compared to pathway 1 physical diffusion, how efficient is pathway 2 involving chemisorption and biotransformation processes? This is the key to ensuring that this approach can be used effectively to inhibit corrosion, please clarify and explain.

(3) Section 2.2 Experimental Models and Materials

I suggest that the authors add a table to present the components of the studied materials. This will allow the reader to compare the differences in their composition.

(4) Section 2.3 Preparation of cup-shaped specimens of concrete

The headings of the whole Section 2 are confusingly numbered Please revise

Figure 3. Please add subheadings A, and B for each figure and distinguish them in the corresponding captions.

(5) 2.4.1. Culture of electricity-producing bacteria isolated from sewer sludge

Only the source of EPB is described here and it is not clear if they are pure bacteria. Plus, what is the process of inoculating SRB described in Lines116-117, please clarify in detail in the text.

(6) Section 2.4 Hydrogen sulfide suppression experiment by conductive concrete.

Please correct the outline number.

Is this test conducted in a sealed environment? Does monitoring hydrogen sulfide concentration refer to gas concentration? How is it detected? Please describe the test procedure in detail.

(7) Section 3.2 Results of adsorption experiment of hydrogen sulfide

Why is OPC not compared in Figure 5? please explain in the text

(8) Section 3.3 Effect of EPB inoculation to H2S reduction

(a)Line 289-291: The authors stated that “The histogram in Figure 6A indicates pH change. At the beginning stage of substrate addition, mostly pH increased to around 9 due to addition of glucose. and then pH of all samples rapidly dropped to approximately ”. Why the description does not match what is shown in Figure 6-A?

(b)The first plot in Figure 6 should be numbered as A

(c)Grammar errors. Rewrite the sentence Lines 291-295, please.

(d) Where does sulfide come from? Is it produced by SRB? If so, how can it be determined that the reduction of H2S is the contribution of EPB and does not be suffered from SRB production efficiency

Author Response

Question 1:

From the introduction section, the practical significance of this study is to reduce the corrosion problem in the wastewater system, however, the causes of corrosion in the wastewater system are not introduced. This aspect is what needs to be added. It is recommended to read the following recent articles (or some others. If needed, please search by yourselves)

(a). Madraszewski S, Dehn F, Gerlach J, et al. Experimentally driven evaluation methods of concrete sewers biodeterioration on laboratory-scale: A critical review[J]. Construction and Building Materials, 2022, 320: 126236.

(b). Wu M, Wang T, Wu K, et al. Microbiologically induced corrosion of concrete in sewer structures: A review of the mechanisms and phenomena[J]. Construction and Building Materials, 2020, 239: 117813.

Response 1: Thank you very much for your comments and your recommendation to recent artcles. Reviewer’s attention is right. It is common knowledge that sulfide species in bulk wastewater formed from sulfate source by sulfate reducing bacteria are the causes of corrosion in the wastewater system. This introduced much in our previous work. However, we added two references to mention this aspect in this resubmission revision.

Question 2:

(2) In Figure 1 and section 2.1 Oxidation of sulfide by electron-emitting,

(a)the biofilm is presented to be below the sediment, which is not quite realistic. There is no clear boundary between the two, anaerobic bacteria are in presence of the sediment. Please redraw it.

Response 2: Thank you very much for your comment. Of course, there is not clear boundary between the biofilm and sediment, this expected to imagine boundary that biofilm was transformed from sludge sediment. In this revised manuscript, we edited it to not be confused to reader at the Figure 1.

Question 3:

(b) Why do electrons pass through concrete when they can be transferred directly through sewage? This is not following the laws of physics, because the resistance of concrete is much higher than that of water. Please clarify and explain this aspect.

Response 3: Thank you very much for your question. Your comments are very interesting. Hydrogen sulfur generated from anaerobic environment in the sludge sediment layer without presence of oxygen. pH elevation and addition of NaOH, Ca (OH)2 or biocides can inhibit H2S generation. That development of alternative electron acceptors as oxygen, nitrate, nitrite... also reduce activities of sulphate reducing bacteria (SRB) will prevent production of H2S. Moreover, activity condition of EPB bacteria on surface concrete at the bottom is relate to H2S control. In this work, there exist three factors, EPB activity, electron pathway of conductive concrete and electron acceptor of oxygen nearby concrete surface. these conditions help it operate as a microbial fuel cell device and this is the significant finding in this study. Of course, electrons can pass through liquid environment of sewage, this is not enough components to create a MFC activity. EPBs release to generate electrons when organic matter and hydrogen sulfide are oxidized, electron transfers from bacteria to electron acceptors, an electron pathway via to conductive concrete to enable biological oxidation process of H2S.  H2S removal occurs in the anodic section of surface of conductive concrete

Question 4:

(c) Line 129, The authors stated that “Therefore, in this present study, the work plan designed assumed that "oxidation of sulfide by electron-releasing bacteria" as shown in Figure 1.”  Why is it based on a hypothesis, for experimental verification? Is it to match the evidence? Please provide relevant supporting research.

Response 4: Thank you very much for your comments. We also revised some explanation at the section of materials and methods in the resubmission manuscript for easy reader. All experiments were conducted to confirm role of conductive concrete in the way to suppress H2S. Hydrogen sulphur generated from anaerobic environment in the sludge sediment layer without presence of oxygen. EPBs release to generate electrons when organic matter and hydrogen sulfide are oxidized, electron transfers from bacteria to electron acceptors, an electron pathway via to conductive concrete by Sun earth in this work to enable biological oxidation process of H2S.  H2S removal occurs in the anodic section of surface of conductive concrete.

Question 5:

(d) After H2S is generated, there are two paths available to it, path 1 is to diffuse into the effluent and have the opportunity to spill over the water level causing concrete corrosion, and path 2 is, as the authors assumed, to be adsorbed onto the concrete surface, meaning that they are converted into elemental sulfur in the sediment. Compared to pathway 1 physical diffusion, how efficient is pathway 2 involving chemisorption and biotransformation processes? This is the key to ensuring that this approach can be used effectively to inhibit corrosion, please clarify and explain.

Response 5: Thank you very much for your comment and question. Compared to pathway 1 physical diffusion as mentioned, pathway 2 of occurrence on concrete surface, conductive material and anode area of  surface oxygen. these factors make it as a microbial fuel cell in H2S suppression and in this study, we concentrated to confirm on three ways: Hydrogen sulfide generated from anaerobic environment in the sludge sediment layer without presence of oxygen. One mechanism to explain for findings in this work is the combination of adsorption capacity of conductive concrete and EPB’s activity releases electrons when organic matter and hydrogen sulfide are biologically oxidized, electron transfers from bacteria to electron acceptors, an electron pathway via to conductive concrete by Sun earth in this work to enable biological oxidation process of H2S.  H2S removal occurs in the anodic section of surface of conductive concrete. We also revised some explanation at the section in the resubmission manuscript.

Question 6:

(3) Section 2.2 Experimental Models and Materials

I suggest that the authors add a table to present the components of the studied materials. This will allow the reader to compare the differences in their composition.

Response 6: Thank you very much for your suggestion. San-earth M5C, Denka Black, Magnetite MTB-30 and OPC used are commercialized cement in Japan for target of conductive concrete in this study. I revised it to easy for reader in the resubmission manuscript. I appreciate your opinion. However, we will refer in further work.

Question 7:

(4) Section 2.3 Preparation of cup-shaped specimens of concrete

The headings of the whole Section 2 are confusingly numbered Please revise

Figure 3. Please add subheadings A, and B for each figure and distinguish them in the corresponding captions.

Response 7:  Thank you very much for your remark. I am sorry for this inconvenience. I added subheadings for A and B in resubmission manuscript at Figure 3.

Question 8:

(5) 2.4.1. Culture of electricity-producing bacteria isolated from sewer sludge

Only the source of EPB is described here and it is not clear if they are pure bacteria. Plus, what is the process of inoculating SRB described in Lines116-117, please clarify in detail in the text.

 Response 8: Thank you for your comment. Reviewer’s confusion is right. Source of EPB isolated and cultivated to become pure bacteria for target experiments. Lines 116-117, "SRB" should be "EPB". I revised it in the resubmission manuscript.

Question 9:

(6) Section 2.4 Hydrogen sulfide suppression experiment by conductive concrete.

Please correct the outline number.

Is this test conducted in a sealed environment? Does monitoring hydrogen sulfide concentration refer to gas concentration? How is it detected? Please describe the test procedure in detail.

Response 9: Thank you very much for your comments. “2.4” should be changed “2.4.1”. I revised it in the resubmission manuscript. Experimental model of cup-shaped specimen as Figure 3A was conducted to assess effect of sulfide reduction. According to this, volume of synthetic wastewater and sludge mixture of excess sludge and digested sludge was prepared, the dissolved oxygen concentration was kept below 0.1 mg/L in anaerobic condition by purging the synthetic wastewater with pure nitrogen bubble as previous work (Imai et al., 2022). The multi-layers of plastic film were covered on water surface to prevent dissolved oxygen to be able to oxidize H2S and diffusion of H2S from inside.

Question 10:

(7) Section 3.2 Results of adsorption experiment of hydrogen sulfide

Why is OPC not compared in Figure 5? please explain in the text

Response 10: Thank you very much for your comment. OPC is not compared in Figure 5 because as referred on Figure 4, electrical resistivity of the concrete in samples OPC, D5, M10, M25 were determined to be highest. Therefore, samples of conductive material San-earth (S50), D10 and M50 with lower electrical resistivity are chosen to make specimens of conductive concrete for next experiments.  

Question 11:

(8) Section 3.3 Effect of EPB inoculation to H2S reduction

(a)Line 289-291: The authors stated that “The histogram in Figure 6A indicates pH change. At the beginning stage of substrate addition, mostly pH increased to around 9 due to addition of glucose. and then pH of all samples rapidly dropped to approximately ”. Why the description does not match what is shown in Figure 6-A?

Response 11: Thank you very much for your constructive comments. I revised them in resubmission manuscript.  

Question 12:

(b)The first plot in Figure 6 should be numbered as A

(c)Grammar errors. Rewrite the sentence Lines 291-295, please.

(d) Where does sulfide come from? Is it produced by SRB? If so, how can it be determined that the reduction of H2S is the contribution of EPB and does not be suffered from SRB production efficiency

Response 12: Thank you very much for your remark and constructive comments. I am sorry it was output mistake. I removed it and replaced in resubmission manuscript. In this time, we revised grammatical mistakes in the resubmission manuscript. All are showed in changes marked. Regarding source of sulfide, under anaerobic condition, sulfate mainly form to sulfide species in wastewater by SRB, and then H2S gas emits in the air phase of sewer pipe system. This sulfide will be transformed to sulfuric acid and react to normal cement materials and cause to corrosion of sewer. EPB community effort to remove H2S and do not be suffered from SRB production.

Reviewer 2 Report

Manuscript Number: ijerph-2092108

Enhancing Biological Oxidation of H2S in Sewer Pipe with Highly Conductive Concrete and electricity-producing bacteria

The authors investigated an interesting topic and used several important techniques to evaluate the experiments also the paper needs significant revision, completion and explanation.

1.      Rewrite the conclusion.

2.      Materials and sample solutions. Give reference.

3.      Add relevant reference articles. Exemple:

https://doi.org/10.1016/j.arabjc.2022.104513

Author Response

Question 1:

The authors investigated an interesting topic and used several important techniques to evaluate the experiments also the paper needs significant revision, completion and explanation.

  1. Rewrite the conclusion.

Response 1: Thank you very much for your constructive comments. In this time, we revised section of conclusion and all manuscript in the resubmission manuscript. All are showed in changes marked.

Question 2:

  1. Materials and sample solutions. Give reference.

Response 2: Thank you very much for your constructive comments. In this time, we revised section of sample solution, references, materials and methods in the resubmission manuscript. All are showed in changes marked.

Question 3:

  1. Add relevant reference articles. Exemple:

https://doi.org/10.1016/j.arabjc.2022.104513

Response 3: Thank you very much for your comments and your recommendation to relevant article. Reviewer’s attention is right. we revised all in the resubmission manuscript. All are showed in changes marked. we added more three references of this aspect in this resubmission revision.

Reviewer 3 Report

Dear authors. This article addresses an important scientific area such as the degradation of corrosive gases, like H2S, that could damage and/or collapse sewage networks. Choosing an original process that combines the biological oxidation in conductive materials with the electricity-producing bacteria. However, this paper requires to be modified to publish it.  The items to improve are:

Writing. English needs to be improved. There are many errors, missing articles, connectors, poor writing, etc. 

Discussion. This section is very short, but it is the fundamental one. It is important to add literature data to compare their results. This also allows this work to have scientific soundness.

Conclusions. It is necessary to report data on the most important parameter of the materials, to improve the final analysis. 

Author Response

Dear authors. This article addresses an important scientific area such as the degradation of corrosive gases, like H2S, that could damage and/or collapse sewage networks. Choosing an original process that combines the biological oxidation in conductive materials with the electricity-producing bacteria. However, this paper requires to be modified to publish it.  The items to improve are:

Question 1:

Writing. English needs to be improved. There are many errors, missing articles, connectors, poor writing, etc. 

Response 1: Thank you very much for your constructive comments. In this time, we revised grammatical mistakes in the resubmission manuscript. All are showed in changes marked. We attached the certificate of English check for this manuscript.

Question 2:

Discussion. This section is very short, but it is the fundamental one. It is important to add literature data to compare their results. This also allows this work to have scientific soundness.

Response 2: Thank you very much for your constructive comments. In this time, we revised section of discussion and all manuscript in the resubmission manuscript

Question 3:

Conclusions. It is necessary to report data on the most important parameter of the materials, to improve the final analysis. 

Response 3: Thank you very much for your constructive comments. we revised it at section of conclusion

Round 2

Reviewer 1 Report

The author responded to most of my comments, but there are some questions whose answers are not yet clear.

Comment 3: Authors stated that “In this work, there exist three factors, EPB activity, electron pathway of conductive concrete and electron acceptor of oxygen nearby concrete surface. these conditions help it operate as a microbial fuel cell device and this is the significant finding in this study. Of course, electrons can pass through liquid environment of sewage, this is not enough components to create a MFC activity. EPBs release to generate electrons when organic matter and hydrogen sulfide are oxidized, electron transfers from bacteria to electron acceptors, an electron pathway via to conductive concrete to enable biological oxidation process of H2S”. However, It is still not clear in following aspects:

What experimental evidence is there for the formation of a MFC between the oxygen electron acceptor and the EPB?

How to determine the electron propagation medium is conductive concrete rather than the effluent which is in contact with concrete?

Comment 9 The authors do not give a clear answer to the question in the text, and I still do not know whether the sulfide concentration was measured in gas or liquid, and how it was detected.

Author Response

Dear Reviewer,

Re: Manuscript ID ijerph-2092108

I appreciated the constructive questions of the reviewer. Regarding to your question and comments, I have addressed each of their concerns as outlined below.

Question 1:

What experimental evidence is there for the formation of a MFC between the oxygen electron acceptor and the EPB? How to determine the electron propagation medium is conductive concrete rather than the effluent which is in contact with concrete?

Response 1:

Thank you very much for your question and comments relating to the formation of a MFC between the oxygen electron acceptor and the EPB. Previous experiment of our work was designed to enable the oxidation/mitigation of sulfide (sulfide ion, hydrogen sulfide ion, and hydrogen sulfide) generated in the anaerobic sludge sediment layer. MFCs is a device that uses electrically catalytic microorganisms to convert the organic or inorganic substance into electricity. The H2S inhibition of conductive material from San-earth (commercial conductive concrete) suggests that an electron pathway may exist between anaerobic to aerobic area as described in below diagram. We use cup-shaped specimen designed same as in this study with a non-conductive epoxy resin inserted surrounding inside concrete material at location of approximately 5 cm below the water surface would result that the epoxy resin prevents the transfer electrons through. Therefore, electrons are not transferred to be close to the surface layer containing dissolved oxygen or in the air, thus the oxidation reaction of H2S should not proceed due to the absence of electron acceptors. Otherwise, if adsorption of conductive carbon is the main mechanism for the suppression of H2S, the effect should not be lost regardless of the insertion of epoxy resin or not.

Experiments using cup-shaped specimens for 3 samples of OPC, San-earth and San-earth with epoxy resin layer were presented to assess the inhibition effect of sulfide ion, a conductive material decorated to the cup-type specimen with the epoxy resin layer was handled. When inserting epoxy resin layer made completely isolated a transferring pathway electron inside conductive concrete by San earth and effectiveness of OPC and San earth with epoxy resin layer are less H2S suppression than cup-specimen of only San earth material. Therefore, the possibility of biological oxidation mechanism has been referred for the formation of a MFC between the oxygen electron acceptor and the EPB. For this revision, we added to conduct this reference at line 171-172, section 2.3 in the resubmission manuscript.

Question 2:

Comment 9 The authors do not give a clear answer to the question in the text, and I still do not know whether the sulfide concentration was measured in gas or liquid, and how it was detected.

Response 2: Thank you very much for your remind. We are sorry for not response completely for this question. The sulfide concentration was measured in liquid.  Sulfide concentrations were measured in this study using the methylene blue method according to USEPA method (376.2). Colour or turbidity in the sample may interfere with the analysis. Sulfide reacts with the reagents to form methylene blue which is measured by colorimeter. The working range of the method is sulfide concentration below 20 mg/L. Colour or turbidity in the samples in this experiment do not interfere with the analysis. We add to clarify this method from line 198-201 at section 2.5 in the resubmission manuscript.

Thank you very much.
